# Effect of Calcium Foliar Spray Technique on Mechanical Properties of Strawberries

**DOI:** 10.3390/plants12132390

**Published:** 2023-06-21

**Authors:** Beata Cieniawska, Piotr Komarnicki, Maciej Samelski, Marek Barć

**Affiliations:** 1Institute of Agricultural Engineering, The Faculty of Life Sciences and Technology, Wrocław University of Environmental and Life Sciences, 50-375 Wrocław, Poland; beata.cieniawska@upwr.edu.pl; 2The Faculty of Life Sciences and Technology, Wrocław University of Environmental and Life Sciences, 50-375 Wrocław, Poland

**Keywords:** spraying parameters, calcium fertilization, mechanical properties, surface pressure, strawberry

## Abstract

The calcium fertilization of strawberry plants (*Fragaria* × *ananassa Duchesne*) was evaluated using two types of nozzles, with two liquid pressure levels and two driving speeds. The calcium content of the leaves and fruit were analyzed via flame photometry. Higher leaf calcium content was found in plots sprayed with standard nozzles, while higher fruit calcium content was observed for those sprayed with air induction nozzles. The fruit quality was assessed by determining the basic physical and mechanical properties, using uniaxial compression tests integrated with surface pressure measurements. Different spraying techniques influenced the mechanical resistance of the fruit. A spraying speed of 5 km/h and an operating pressure of 0.4 MPa significantly increased the firmness of the fruit by ~66%, the critical load level by 36%, and the maximum surface pressure by up to 38%, but did not increase the geometrical parameters of the strawberries. Regular foliar feeding during harvest could improve the mechanical strength of strawberries. An appropriate spraying technique with a calcium agent could effectively improve the mechanical properties of the delicate fruit, which is particularly important for limiting losses during harvesting, transportation, and storage.

## 1. Introduction

Strawberry (*Fragaria* × *ananassa Duchesne*) is one of the most popular berry crops in Poland. In 2021, the area of strawberry cultivation was 33.9 thousand hectares (ha), accounting for almost 25% of all berry crops. The yield reached 48.05 dt/ha, and the harvest reached 162.9 thousand tons. In 2021, Poland was the tenth-largest producer of strawberries globally and the second-largest in the European Union, according to the Food and Agriculture Organisation [1]. High-quality and high-quantity yields require the use of plant protection products, micronutrients, and macronutrients. Chemical plant protection agents, according to the principles of integrated pest management, can be used after agrophages exceed their thresholds of economic harmfulness. However, macronutrients and micronutrients are provided through both the soil and the foliage. Macronutrients include nitrogen, potassium, phosphorus, magnesium, sulfur, and calcium. The key micronutrients include zinc, manganese, molybdenum, copper, iron, and boron. Ensuring the appropriate nutrient amount is important to guarantee high-quality strawberry [2], wheat [3,4], and blueberry [5] crops.

The plant protection and fertilization techniques must deliver the substances to the right places, and in the correct amounts and forms [6]. The treatment must also be of high quality and effectiveness, while limiting the negative impact of agents on the environment, operators, and bystanders. To achieve the intended effects of the treatment, scientists must consider the complexity of the application and the need for appropriate technical and technological spraying parameters. They must also use the right nozzles for administering the treatment, considering the properties of the liquid and atmospheric conditions [7,8,9].

The quality of the application is evaluated based on the following indicators: unevenness of the distribution and precipitation of the liquid, degree of coverage of the sprayed objects, and deposition of the liquid [10,11,12]. The presented results highlight the direct impact of the application method intended to protect the plant on the effectiveness of the treatment. Considering the “Farm to Fork” strategy published by the European Commission (2020), as described in Ref. [13], the use of pesticides must be reduced by 50% by 2030. Therefore, it is important to create treatment scenarios.

To improve the postharvest mechanical strength of the fruit, regular foliar feeding with calcium can be applied during harvesting. Calcium binds to lignin structures in the cell wall, making it thicker, tougher, and more flexible. Benefits of Ca for maintaining the quality and extending the shelf life of fruit have been documented [14,15].

Strawberries are among the most sensitive fruits, with short shelf lives and low mechanical resistance [16]. They quickly lose firmness during ripening, which contributes significantly to their short postharvest shelf lives. Nagata et al. [17] found that subjecting the fruit to a compressive force of >2 N bruised the strawberries’ surfaces after just two days. Additionally, damage from harvesting and mechanical handling result in increased fruit spoilage and loss of supply chain quality [18,19]. Mechanical damage to the fruit is a defect in the biomaterial, and is closely related to the mechanics of the fruit [20]. The fruit texture is influenced by cell turgor and the structure and composition of cell wall polysaccharides [21].

Recently, nondestructive and noninvasive analytical methods have been used to assess fruit quality in order to avoid the complexity, time requirements, and low throughput of destructive methods [22]. Various nondestructive techniques have been introduced, including colorimetry, visible range imaging, visible and near-infrared spectroscopy, hyperspectral imaging, multispectral imaging, fluorescence imaging, the acoustic pulse technique, the electrical pulse technique, and magnetic resonance imaging [23]. Amoriello et al. [24] used neural networks to predict the quality-related traits of strawberries. Studies have shown that positive effects on the mechanical properties of fruit can be achieved by pre-exposure to ultraviolet irradiation, pulsed light, and pulsed electric fields [25,26].

Fruit quality characteristics are still often determined by destructive testing, which guarantees clear and reliable information, but has some limitations due to the time required for the analysis [27]. The most common methods for assessing quality and mechanical properties are puncturing the fruit with a penetrometer (for measuring firmness) or performing compression tests [28,29]. Firmness is a parameter that is influenced by numerous factors, including the internal structure and chemical composition of the fruit [30,31,32]. Komarnicki and Kuta [33] used compression tests to evaluate and compare the maximum forces and destructive pressures to dangerous loads generated when strawberries were picked at different positions. An et al. [34], on the other hand, analyzed the mechanics of strawberry tissue textures at six temperature levels and developed a dynamic finite element model to investigate the susceptibility of strawberries to impact damage. An increased understanding of fruit biomechanics and spraying techniques is crucial for obtaining high-quality strawberries during the postharvest period [35,36,37]. Together with optimal spray types and fertilization parameters, this can rationally minimize the undesirable side effects of agrochemicals.

Therefore, this study aimed to determine the influence of selected technical and technological foliar fertilization parameters on the mechanical properties of strawberry fruit as quality discriminators.

## 2. Results

### 2.1. Degree of Coverage of Sprayed Surfaces

The results of the tests are shown in the diagrams (Figure 1 and Figure 2). No traces of coverage were recorded on the bottom horizontal surfaces, and, therefore, are not presented. For the upper horizontal surfaces, higher coverage values were obtained with the standard nozzles, except when sprayed at 400 kPa with a driving speed of 10 km/h on 30 May 2022 (Figure 1). Notably, the smallest differences in coverage and highest standard deviation (SD) values were obtained on 30 May 2022, which corresponded to unfavorable weather conditions. The other three measurement days produced clearly higher values for the standard nozzles, with the highest coverages obtained at a pressure of 400 kPa and driving speed of 5 km/h, confirming the previous results.

The differences in the vertical surface coverage (Figure 2) were smaller than those of the horizontal surfaces. The lowest coverage values were observed for the air induction nozzles with driving speeds of 10 km/h and 200 kPa pressure, except on 30 May 2022. As with the upper horizontal surfaces, the atmospheric conditions, particularly the wind speeds, determined the nature of the fertilization process. Higher coverage values were recorded for the standard nozzles, except for during the treatment performed on 6 June 2022 at 5 km/h.

The results of the degree of coverage were subjected to statistical analysis. The F test was used to assess the significance of the results at the significance level of *p* = 0.05. The analyses were performed separately for surface area, pressure, and driving speed. Based on the results of the analysis of variance, which are shown in Table 1, it can be concluded that all analyzed factors had an impact on the degree of coverage, at a significance level of *p* = 0.05.

### 2.2. Deposition of the Spray Liquid

The spray application test results are shown in Figure 3. Higher foliar calcium contents were obtained in the plots sprayed with the standard XR 11003 single-stream nozzles, except when spraying with a pressure of 200 kPa and a forward speed of 10 km/h. The calcium content in the fruit showed different results. In this case, higher calcium contents were obtained for the single-stream air induction nozzle AIXR 11003, except when sprayed with a pressure of 400 kPa and a forward speed of 10 km/h.

### 2.3. Characteristics of the Research Material

Table 2 shows the basic physical properties of the selected test material. Comparing the control group (group 9) with groups 1–8 for different spraying parameters showed an effect of the driving speed on the fruit mass and diameter (*p* < 0.05). Compared with the control group (group 9), slightly lower weights and diameters were measured for the fruits that were sprayed at 5 km/h. The mean weights and diameters were 10.2 g and 27.1 mm, respectively.

The changes in the firmness of the strawberry fruit as functions of the spray parameters are presented in Figure 4. The results for the control group also showed a significant effect of the spray speed (v5), which was 5 km/h (*p* < 0.05).

The highest firmness value, 5 N, was obtained for fruit sprayed the most intensively, that is, at a speed of 5 km/h and an operating pressure of 0.4 MPa. The measured firmness showed the consumptive maturity of the fruit. The differences in firmness between the testing and control groups were approximately 66%. Fruit sprayed at a higher speed (v10; 10 km/h) showed no significant differences in firmness (*p* > 0.05). These results (Table 2, Figure 4) indicate that reducing the spraying speed to 5 km/h at 0.4 MPa clearly increases the firmness, but not the geometric parameters, of strawberry fruit at a spraying speed of 10 km/h.

### 2.4. Destructive Tests

Figure 5 shows the effect of the spray parameters on floc variation (bioyield point) in terms of limiting load-induced permanent deformation. Statistical analysis in the form of the Shapiro–Wilk test showed that all groups conformed to a normal distribution, with the F–test confirming the variance in homogeneity.

With respect to the control group (1), the most significant effects on the load change (*p* < 0.05) were seen in groups 3 and 4 when sprayed at a speed of 5 km/h and operating pressure of 0.4 MPa. These groups showed the greatest mechanical resistance, with the forces increasing by 36.2% (12.9 N) and 16.4% (9.8 N), respectively. In contrast, an intergroup analysis showed the effect of working pressure on groups 1 and 3 (*p* < 0.05, correlation coefficient: 0.44) at peak loads of 9.6 N and 12.9 N, respectively. The compression test results confirmed that groups 5 and 6, when sprayed at a speed of 10 km·h^−1^ and a pressure of 0.2 MPa, were the least affected in terms of load increase (the load increase was just over 2%). The floc forces correlated noticeably with the calcium content of the fruit (Figure 3) and the degree of coverage of the applied product (Figure 1 and Figure 2).

### 2.5. Measurements of Surface Pressure

Figure 6 shows a section of an example curve, F = f(ΔL), with the contour images, illustrating the process of determining surface pressure changes in a compression test. The delicate anatomical structure of the strawberry fruit, which consists of achene seeds on the skin surface with a very soft or hollow core, greatly influences the mechanical resistance. In general, fruit compression can be divided into four phases: the initial contact phase, compression proper, maximum deformation, and tempering.

Visualization of the surface pressures showed that increasing the load gradually increased the contact area up to a maximum value. Beyond this, the pressure dissipated due to the incomplete core and filling of the free cellular spaces, and the fruit underwent permanent inward deformation, resulting in a decrease in load and failure. Based on the compression curve analyses, the critical deformation ranges of all the tested strawberry groups were approximately 7–8.1 mm. The destruction of the fruit was also compounded by the presence of the achene seeds, which, upon exceeding the limiting force *F*_loc_ due to their hardness, damage the skin and penetrate the flesh. The surface pressure images for groups 1–8 were analyzed, and the maximum (peak) values of Pmax were determined, which occurred when the floc force reached its limiting value and was concentrated on the seeds. Figure 7 presents the effect of the spray parameters on the changes in the maximum surface pressures recorded during the compression tests.

The control group clearly exhibited the lowest level of maximum surface pressure value (90.5 kPa), indicating the poor mechanical properties of the fruit. Statistical analyses showed that the surface pressure values of only groups 5 and 6 (for v10 and p0.2) did not differ significantly from those of the control group (*p* > 0.05), as their maximum pressure values increased by 5.4% and 15.4% (95.4 and 104.5 kPa), respectively. In the other cases, product application increased the maximum pressure from approximately 120 kPa to 147 kPa, which was particularly noticeable in fruit sprayed at an operating speed of 5 km·h^−1^. A relation was observed between groups 5 and 7 and groups 6 and 8, showing a significant effect of the spray pressure (*p* < 0.05; correlation coefficient = 0.49). We found no significant differences in the effect of the nozzles on the maximum surface pressure (*p* > 0.05). Comparing the pressure values with the field and chemical test results also revealed a similar pattern of changes in the degree of coverage and calcium content of the fruit. The application of calcium sprays in each case resulted in improved strawberry fruit quality and increased mechanical resistance.

## 3. Discussion

Herein, we assessed the spray parameters as quality indicators. Many studies have determined coverage and application rates using different technical and technological spraying parameters and nozzle types in field trials. Similarly to our study, Etheridge et al. [38] used single-stream standard and ejector nozzles to spray their selected herbicides. They concluded that the treatment’s efficacy was not influenced by the nozzle type. Conversely, Creech et al. [6] observed that the greatest coverage in a maize canopy was obtained with standard single-stream nozzles. They conducted their experiments using Mylar^TM^ cards as samplers. Considering the samplers which were used, Fergusson et al. [39] demonstrated a strong correlation between the results obtained after analyzing WSPs and Mylar^TM^ cards.

Sharpe et al. [40] confirmed that the degree of coverage of strawberries increased with increasing liquid volumes, similarly to our study. Considering the volume of liquid applied, Wójcik and Lewandowski [41] proved that calcium application increased its content in the leaves and fruit, which was confirmed by the results of this study. Furthermore, a study of the calcium content in plants was performed by Ismail and Ganzour [42]. Their experiments used moringa leaf extract at different concentrations, both with and without the addition of 2% potassium nitrate (KNO_3_). A mixture of 6% of the extract and 2% of KNO_3_ increased the yields and fruit characteristics. This agreed with the findings of Hassan [43] and Hamail et al. [44]. In contrast, Husein and Al-Doori [45] evaluated the effects of boron and calcium spraying on the quality and yield of strawberries. They used four experimental combinations: control, 100 mg/L calcium, 20 mg/L boron, and 100 mg/L calcium with 20 mg/L boron. Spraying with 100 mg/L calcium produced the highest mean fruit length and diameter, mean fruit size and weight, mean yield per plant, and yield per area, while spraying 100 mg/L calcium with 20 mg/L boron produced the most flowers and fruit per plant. Mohamed et al. [46] highlighted the importance of macronutrients and micronutrients for plants’ growth, yield, and fruit quality. The results of these previous studies clearly indicate a change in the mechanical properties of strawberry fruit, depending on the spraying technique and the response to the Ca agent.

Firmness is the primary indicator for assessing strawberry flesh quality. An analysis of firmness showed variation in the results compared to those of other studies due to the different cultivars, maturity stages, and research methods. In a study of the foliar application of calcium and boron on strawberries of the cultivar “Chandler,” Singh [47] also showed an absence of weight gain and increased fruit firmness. Previous studies typically used hand-held penetrometers with different pin sizes [24]. A similar trend was observed by Sidhu et al. [48], who showed that spraying with calcium nitrate significantly improved the firmness of the “Winter Dawn” cultivar by imparting strength and thickness to the fruit cell walls (by 2.2 N), using a texture analyzer with a 5-mm diameter tip. Firmness values similar to those of the control group (Figure 7) at 3 N were observed by Wei et al. [49] for the cultivar “Hongyan” after strawberries were sprayed with tea tree oil. Soppelsa et al. [50] found that chitosan increased the firmness of the flesh by approximately 20%, with potential implications for extending the shelf lives of treated fruit. Similar results were also obtained by Bhaskara Reddy et al. [51] for different strawberry cultivars.

Some studies have suggested that chitosan and plant probiotic bacteria can be used as natural bioregulators for safe strawberry fruit production [52,53]. This was confirmed by Hernández-Muñoz et al. [21], who showed that adding calcium to a 1% chitosan solution increased the fruit firmness. Ferreira et al. [54] subjected strawberries to continuous compression at different temperatures and found that the fruits subjected to hydrocooling at 1 °C were more resistant to compression, while those at higher temperatures (20 °C or 24 °C) were more resistant to impact. Duarte-Molina et al. [55] assessed the mechanical properties of strawberries exposed to different doses of pulsed light at 6 °C, and demonstrated cell wall strengthening and significant subcutaneous cell wall integrity induced by pulsed light stress. In puncture tests with a mandrel 4.8 mm in diameter, the fruits achieved force and displacement values similar to those obtained in this study. Load studies, as a function of percentage deformation for less ripe strawberries, were also conducted by An et al. [28], who, in uniaxial–radial compression tests at higher deformation rates (1–5 mm/s), obtained slightly higher peak loads of approximately 20 N. Regarding strength testing of strawberry fruit, very few studies have reported surface pressure measurements. Compression tests, in relation to surface pressure measurements, were conducted by Komarnicki and Kuta [33]. They determined the effect of the working position on changes in the load, as well as on the picker’s musculoskeletal system and the surface pressure exerted on the fruit during hand picking, which determined the preservation of the strawberry fruit’s quality. For a deformation speed of 10 mm/min, they observed destructive forces and maximum surface pressures comparable with those of this study.

The work showed that the improvement in fruit firmness was largely due to the adopted spraying technique (especially the lower speed and higher pressure of the treatment). However, based on the degree of coverage and the chemical and strength tests, it was proven that an effective increase in the value of the measured parameters would not be possible without the application of foliar spraying with a calcium agent.

## 4. Materials and Methods

### 4.1. Experimental Set-Up

This study was conducted on a Sibilla cultivar strawberry plantation in the village of Jaszkotle in the municipality of Kąty Wrocławskie, Poland (51°03′22.3″ N, 16°54′03.5″ E). Strawberry (“Sibilla”) is a Italian, medium-ripening cultivar with high production potential. The fruits are bright red in color. This cultivar shows high adaptability, both in the open field and under covers. “Sibilla” is a cultivar which is suitable for the continental European climate.

The experiments were conducted on two 50 m rows in 10 m sections. Each row was 1.0 m wide, and the spacing between rows was 0.5 m. Fertilization was performed four times, on 30.05, 6.06, 14.06, and 22.06. Fertilization treatments were carried out starting from the beginning of the fruit ripening process (81st phenological stage on the BBCH scale). The beginning of each section included a marker describing the parameters of the spraying device. While treating a particular section, the adjacent sections were protected from the spray with an agrofleece. The spraying device was a nozzle carrier with a self-propelled sprayer functionality. The device comprised a liquid system and a driving system (Figure 8).

The liquid system consisted of a tank, pump, control valve, pressure gauge, and four multinozzle bodies with mounted nozzles. It was meant to achieve and maintain a set pressure. The chassis system allowed the nozzle carrier to move, and was equipped with an electric motor, running wheels, and a chain. The processes of starting and stopping the spraying unit, changing the traveling speed, and switching the nozzles on and off were performed using a control panel (Figure 9). The speed of the carrier was changed using a frequency converter. Speeds of 5 km/h and 10 km/h corresponded to frequencies of 13 Hz and 17 Hz, respectively.

Fertilization was performed using a calcium-containing agent from Cosmocel at a dose of 1 L of agent per 500 L of water. The fertilization parameters utilized were as follows: travel speeds of 5 km/h and 10 km/h, pressures of 200 kPa and 400 kPa, single-stream standard nozzle XR, air induction nozzle AIXR, and a boom height of 0.5 m. The characteristics of the nozzles which were utilized are described in Table 3.

The layout of the experimental plots and the spray parameters applied to each section are presented in Figure 10.

Before each run, the atmospheric conditions were measured and the nozzle flow rates were monitored. The atmospheric conditions were measured using an anemometer equipped with a thermometer and a humidity sensor. The wind speed, temperature, and humidity measurements are summarized in Table 4.

The control of the flow rate of the liquid was monitored for the nozzles which were used. To determine the flow rate, the liquid was collected for 1 min in measuring cups placed under the nozzles. The quantity of the collected liquid was then read and compared with the value provided by the nozzle manufacturer. For any differences between the measured value and the catalog value, the liquid pressure was corrected.

### 4.2. Degree of Coverage of Sprayed Surfaces

The degrees of coverage of the sprayed surfaces were analyzed using water-sensitive paper samplers (WSP; 76 mm × 26 mm; Syngenta Crop Protection AG, Basel, Switzerland). The samplers were attached to tripods to create vertical and horizontal surfaces (Figure 11).

Three tripods were set up in each test plot, allowing the experiment to be performed in triplicate. The first tripod was set up 3 m from the start of the plot, with the remaining two set up at 2 m intervals from each other. After spraying, the WSPs were removed; attached to previously prepared stencils; and, once dry, protected from moisture. They were then scanned and analyzed using Adobe Photoshop (22.0) 2021 (Adobe Inc., San Jose, CA, USA). During the analysis, three sections measuring 1 cm^2^ were randomly selected on the WSP surfaces. The pixel counts for the 1 cm^2^ samples were read from the corresponding histograms before the areas of liquid coverage were marked on the same surface and the number of pixels was read. The degree of coverage was calculated as the ratio of the number of pixels on the liquid-covered area to the number of pixels on the entire analyzed area (Formula (1)).
(1)$$P_{sp}=\frac{P_{pc}}{P_{a}}\cdot 100[\%]$$
where *P_sp_* is the percentage degree of coverage, *P_pc_* the liquid-covered area in pixels, and *P_a_* is the test area in pixels.

### 4.3. Deposition of the Sprayed Liquid

Chemical analyses were performed at the Department of Horticulture of the Wrocław University of Environmental and Life Sciences. The calcium contents of the leaves and fruits were measured by flame photometry (Jena Model III, Zeiss, Poznań, Poland). In brief, 0.4 g of dry, ground material was weighed and sieved through a 1 mm mesh. The extraction solution consisted of 2% chda acetic acid. The sample was placed on a laboratory shaker for 30 min at a shaking frequency of 150 rpm/min. The resulting suspension was filtered on a quantitative medium filter (Chemland). After filtration, a flame photometry reading was taken. Simultaneously, calcium carbonate standard solutions of known concentrations (100, 200, 400, and 500 mg/dm^3^) were prepared and used to obtain a standard curve. The soluble calcium content of the sample was measured and converted to mg/100 g^−1^ dry matter as the dry samples were added by weight.

The parameters of the FA flame photometer were as follows: air pressure, 30–40 kPa; acetylene pressure, 80–90 mm of water; flame height, approximately 10 cm; height of the flame cone (on the matrix), approximately 3 cm; atomizer nozzle cross section, 0.6 mm; and Ca filter, 63 J.

Leaf and fruit samples were taken twice, on 14 June 2022 and 27 June 2022. In addition, leaves were collected on 29 May 2022 to measure the calcium content prior to fertilization. The research samples from each plot included 10 leaves and 10 fruits.

### 4.4. Laboratory Evaluation of Mechanical Properties

In the final stage, the physical and mechanical properties were determined in order to assess the quality of the strawberries. The freshly picked fruit were transported to the agrophysics laboratory of the Institute of Agricultural Engineering, where they were selected based on their mass, geometric size, and harvest maturity. The laboratory temperature was 28 °C ± 1 °C, with a relative humidity of 55%. Individual fruit was weighed using an electronic balance (AXIS, AD500, Puszczykowo, Poland) with a 500 g range and a 0.001 g accuracy. The average fruit diameter was determined using an electronic caliper with an accuracy of 0.01 mm (Hogetex, Varsseveld, The Netherlands). The harvest maturity was determined for 90 selected fruits (9 groups of 10 replicates) via firmness tests, using a digital fruit firmness penetrometer (GY–4; Newtry, Huizhou, GuangDong, China) with an accuracy of 0.01 N and a stem diameter of 3.5 mm, designed for soft fruit. The penetrometer was mounted on a lever handle to ensure repeatable travel conditions. For each selected group, whole-fruit compression tests were performed on an Instron 5566 machine (Norwood, MA, USA) integrated with a surface pressure system from Tekscan (South Boston, MA, 02127, USA) (Figure 12).

The data were transmitted to the computer via a multichannel hub (VersaTek 8-Port Hub) connected directly to a sensor handle (VersaTek Sensor Handles), which contained a pressure sensor. The system, together with the I-Scan software, allowed for real-time data recording with a sampling rate of up to 5 kHz. To avoid disturbing the internal structure of the fruit, measurements were taken on the strawberries and stalks. The fruits were laterally placed on a film pressure sensor (model 5051, range 3447 kPa) fixed on a nondeformable substrate, and were subjected to uniaxial compression between two parallel plates until failure at a head loading rate of 10 mm/min. This enabled the simultaneous monitoring of failure loads, deformation, contact area, and maximum surface pressures, which were determined from contour images. In total, 180 fruits were tested, with the experiments repeated 20 times for each of the 9 groups tested.

### 4.5. Statistical Analysis

Basic statistical analyses were performed using Microsoft Excel. A one-way analysis of variance (ANOVA) was performed using GRETL (Allin Cottrell, Wake Forest University, Winston-Salem, NC, USA) to determine significant differences between the dependent strength parameters and independent variable spray parameters. The correlation coefficients were also determined between the spray parameters (nozzle type, driving speed, and pressure) and strength parameters (firmness, local breaking load, and maximum surface pressure). Coverage rate tests were performed in triplicate, with three sections randomly selected on the WSP surfaces. The coverage test results were analyzed using a multivariate ANOVA in Statistica (version 13.1; Tibco Software, Palo Alto, CA, USA). Statistical significance was determined using F-tests, and was set at *p* = 0.05.

## 5. Conclusions

Studies evaluating the quality of strawberries of the *Sibilla* cultivar after foliar calcium feeding showed an overall improvement in both the physical and mechanical properties of the fruit. This article confirms that different spraying techniques influence the mechanical resistance of the fruit. The working speed and spray pressure were key parameters determining the application efficiency. Compared with the control group, a spray rate of 5 km/h and an operating pressure of 0.4 MPa significantly increased the fruit’s firmness (up to 66%), critical load level (up to 36%), and maximum surface pressure value (up to 38%), without changing the geometric parameters of the strawberries. The evaluation of the surface pressures of strawberries should consider the anatomical structures and the destructive effect of the hard seeds of the achenes, which can be observed using foil pressure sensors. Studies have shown that an appropriate calcium spray technique can effectively improve the mechanical properties of the delicate strawberry fruit, which is crucial for reducing losses during its harvesting, transportation, and storage.

## Figures and Tables

**Figure 1 plants-12-02390-f001:**
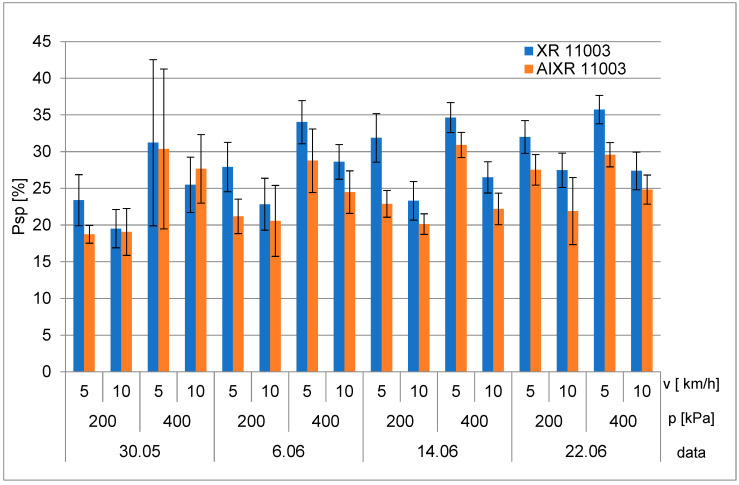
The degree of coverage of horizontal top surfaces. Error bars indicate mean ± SD.

**Figure 2 plants-12-02390-f002:**
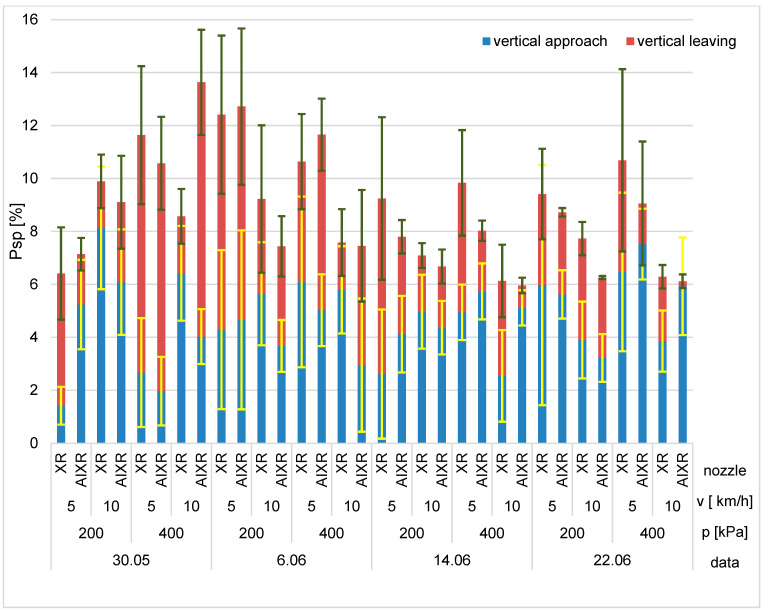
The degree of coverage of vertical surfaces. Error bars indicate mean ± SD.

**Figure 3 plants-12-02390-f003:**
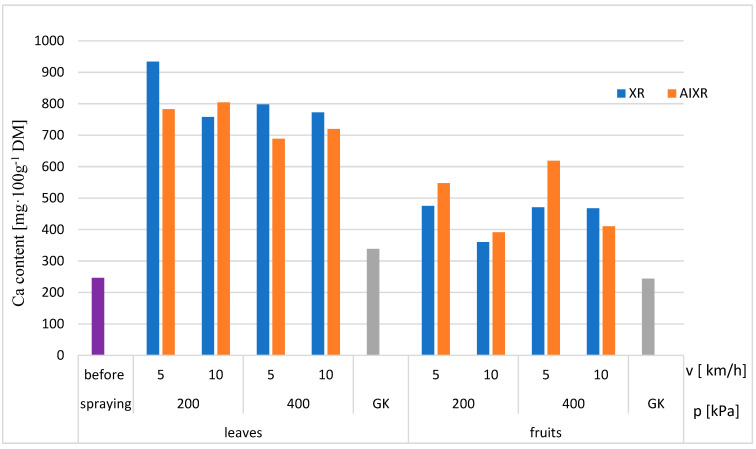
Calcium contents of leaves and fruits.

**Figure 4 plants-12-02390-f004:**
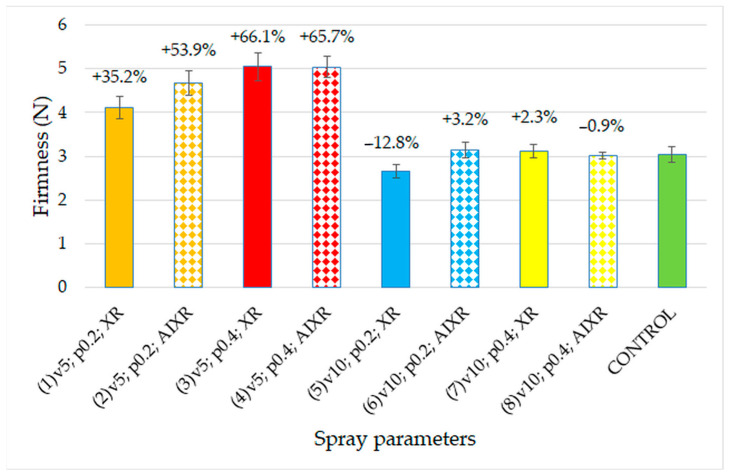
Changes in fruit firmness in relation to spray parameters. The designation of spray parameters according to Table 2. Error bars indicate mean value ± SE. Groups 1–4 are statistically significant (*p* < 0.05).

**Figure 5 plants-12-02390-f005:**
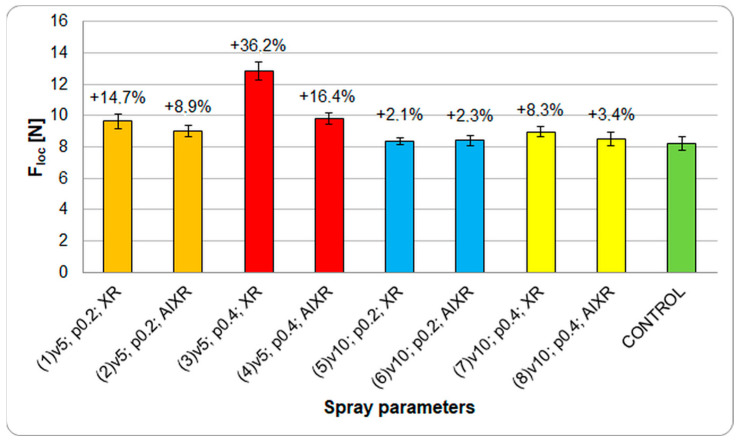
The effect of spray parameters on changes in critical fruit loads during compression tests. Spray parameters are designated according to Table 2. The percentage changes represent changes with respect to the control group. Error bars represent the mean ± standard error (SE).

**Figure 6 plants-12-02390-f006:**
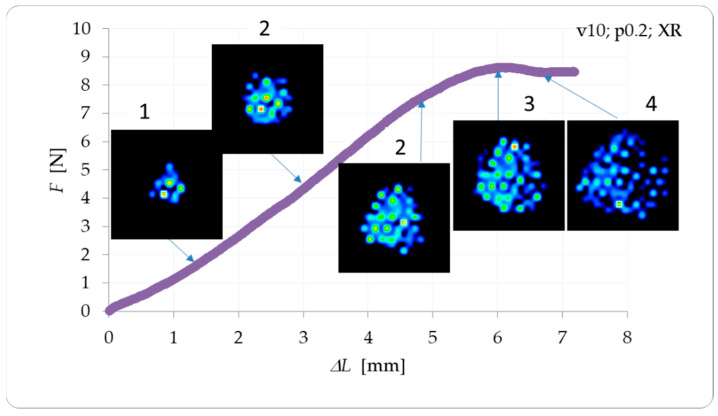
An example of a compression curve with an illustration of surface pressure contour changes.

**Figure 7 plants-12-02390-f007:**
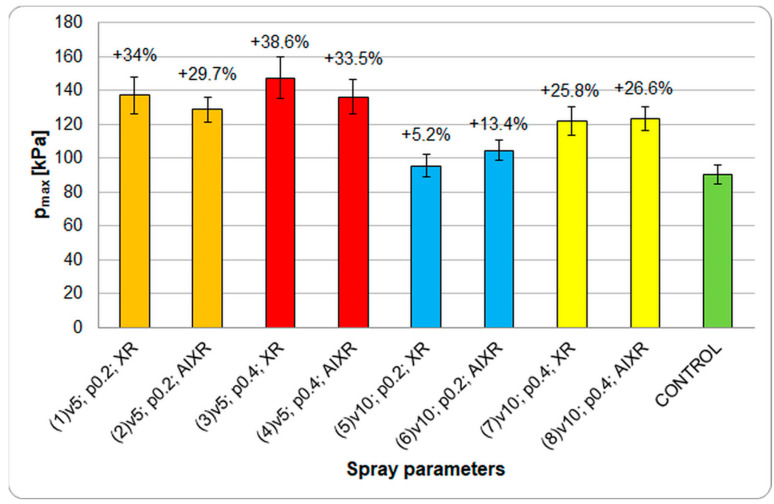
The effect of the spray parameters on changes in the maximum surface pressure during the compression tests. The spray parameter designations are the same as those in Table 2. Percentage changes represent changes with respect to the control group. Error bars represent the means ± SE.

**Figure 8 plants-12-02390-f008:**
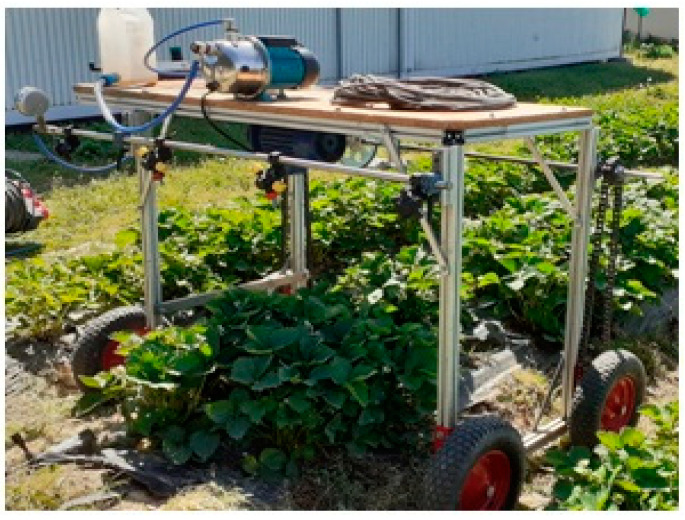
View of the spraying device.

**Figure 9 plants-12-02390-f009:**
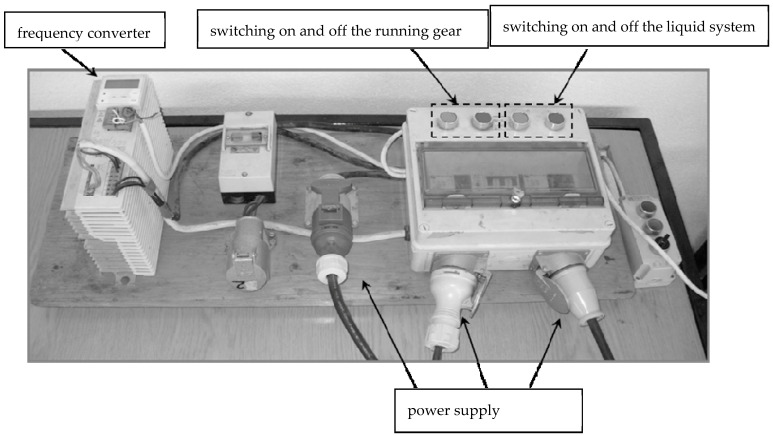
Sprayer control panel.

**Figure 10 plants-12-02390-f010:**
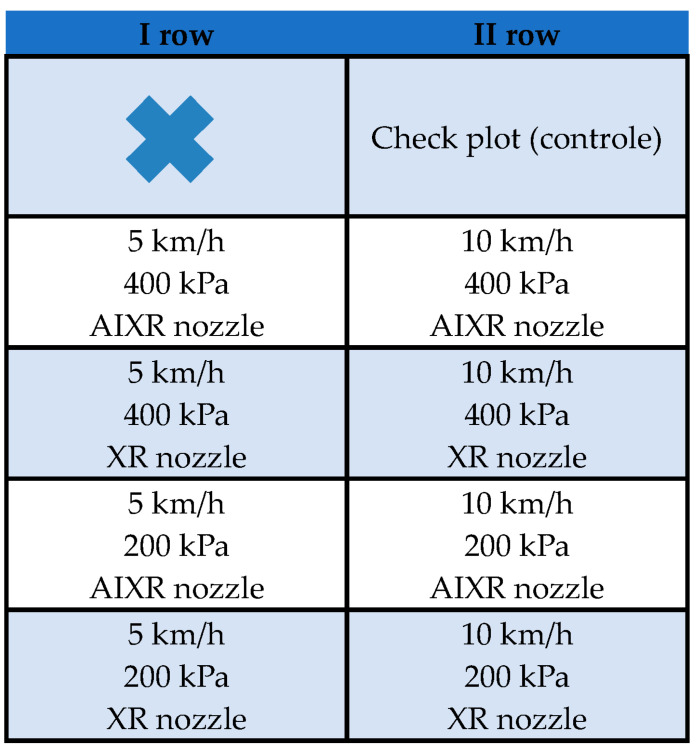
Layout of the test plots.

**Figure 11 plants-12-02390-f011:**
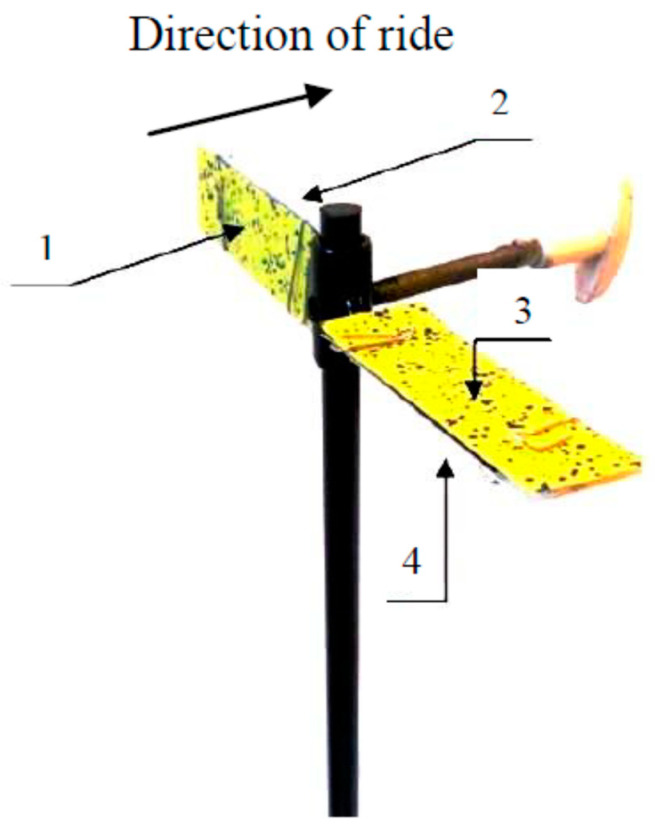
Water-sensitive paper (WSP) positioning on each tripod: 1, vertical approach surface; 2, vertical leaving surface; 3, horizontal upper surface; and 4, horizontal bottom surface.

**Figure 12 plants-12-02390-f012:**
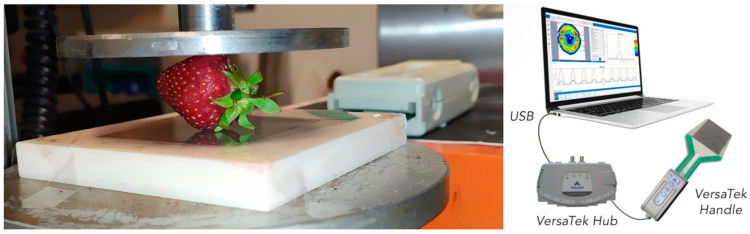
View of the compression test with a surface pressure measurement system (developed from www.tekscan.com (accessed on 29 May 2023)).

**Table 1 plants-12-02390-t001:** The analysis of variance.

Date of Fertilization	Factor	F	*p*
30.05	Pressure	39.78	<0.005
	Driving speed	3.26	0.043
	Surface	77.71	<0.005
6.06	Pressure	4.3	0.016
	Driving speed	22.3	<0.005
	Surface	142.4	<0.005
14.06	Pressure	33.7	<0.005
	Driving speed	100.5	<0.005
	Surface	526.6	<0.005
22.06	Pressure	19.8	<0.005
	Driving speed	49	<0.005
	Surface	388	<0.005

**Table 2 plants-12-02390-t002:** Basic physical properties of the test material.

Number of Groups	Spray Parameters	Mass (g)	Diameter (mm)	Homogeneity Groups
1	v5; p0.2; XR	8.48 ± 0.49	25.51 ± 1.13	xxx
2	v5; p0.2; AIXR	9.83 ± 1.67	26.68 ± 1.21	(-)
3	v5; p0.4; XR	8.98 ± 0.92	25.77 ± 1.27	xxx
4	v5; p0.4; AIXR	9.85 ± 1.14	26.48 ± 1.68	(-)
5	v10; p0.2; XR	10.5 ± 0.9	27.51 ± 1.82	(-)
6	v10; p0.2; AIXR	11.53 ± 1.13	28.78 ± 1.36	(-)
7	v10; p0.4; XR	12.74 ± 2.14	29.77 ± 1.81	(-)
8	v10; p0.4; AIXR	9.81 ± 0.8	26.38 ± 1.15	(-)
9	CONTROL	10.22 ± 1.37	27.11 ± 1.49	
	MEAN	10.22 ± 1.17	27.11 ± 1.44	

The weights and diameters are presented as their means ± SD. The spray parameters were as follows: v5 and v10 (travel speeds of 5 and 10 km/h, respectively), p0.2 and p0.4 (spray pressures of 0.2 and 0.4 MPa, respectively), and nozzle types XR and AIXR (standard and air induction nozzles, respectively). Each of the nine groups comprised ten replicates. Marking (xxx) indicates significant differences between groups 1–8 and the control group (*p* < 0.05), as well as homogeneous groups (-).

**Table 3 plants-12-02390-t003:** Nozzle characteristics (Source: TeeJet Catalogue 52–M, 2023).

Name of Nozzle	Name of Manufacturer	Type of Nozzle	Size of Nozzle	Spray Angle	Flow Rate at Pressure 200 kPa (L/min)	Flow Rate at Pressure 400 kPa (L/min)
XR	TeeJet, Spraying Systems Co.	Standard	03	110°	0.96	1.36
AIXR	TeeJet, Spraying Systems Co.	Air induction	03	110°	0.96	1.36

**Table 4 plants-12-02390-t004:** Atmospheric conditions measured during the study.

Day	Temperature (°C)	Wind Speed (m/s)	Air Humidity (%)
30 May 2022	17	2	77
6 June 2022	21	0.5–0.6	65
14 June 2022	23	0.5–0.6	62
22 June 2022	23	0.3	62

## Data Availability

Not applicable.

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
