# Peer review of "Effect of Calcium Foliar Spray Technique on Mechanical Properties of Strawberries"

_plants, 2023, doi:10.3390/plants12132390_

Round 1
Reviewer 1 Report
The research work is very well-planned and performed, the chosen system is adequate for this research. The methods were described in detail, so the experiments can be repeated by others and the presentation of the results is clear, the figures and tables are well-prepared. In general, the manuscript is written in a good manner and order. However, it raises a question in methodology:
Were the leaves and fruits surface washed during sampling before the determination of calcium content? Because we are interested in the calcium content that the plants really build in.
Author Response
Response to Reviewer 1
Dear Reviewer,
Thank you for your comments and suggests concerning our manuscript entitled “Evaluation of strawberry fruit quality in terms of foliar calcium fertilization” (plants-2455513). Those comments are all valuable and very helpful for revising and improving our paper, as well as the important guiding significance to our researches. We have studied comments carefully and have made correction which we hope meet with approval. Revised portion are marked in red in the paper. The main corrections in the paper and the responds to the reviewer’s comments are as following.
Point 1. Were the leaves and fruits surface washed during sampling before the determination of calcium content? Because we are interested in the calcium content that the plants really build in.
Response 1: The leaves and fruits were not washed before the determination of calcium content.

Reviewer 2 Report
The manuscript titled „Evaluation of Strawberry Fruit Quality in terms of Foliar Calcium Fertilization” is about an actual question. The Ca-supply is very important for the strawberries, and there is a big stress on the spraying techniques.
I think the introduction should contain some data about the optimal Ca content of fruits and leaves.
The „2. Results” and the third sub-cheapter („3. Results”) marked with the same sub-title, Please change the „2. Results” to „4. Materials and methods”. Correct order of the sub-chapteres for MDPI Plants is abstract, introduction, results, methods and materials, conclutions, references. Please douple check order of sub-chapters in your paper.
I think, aim of a paper is to provide data and results, which are repeatable, therefore it is very important to write the Materials and methods prefectly. So, it should be provided more detailed information about the orchard desing (e.g. Was it one row, or more rows in the platation?, How wide was the bed? Were the plants planted on burm? How was the density or spacing in the examined orchard?, When was the orchard established? In which leaves were the plants? What kind of soil had the orchards? How was the Ca-supply of the soil?)
In the strawberry production the cultivars change very frequently, therefore it would be good to add a very short description about Sybilla cultivar. Is this cultivar an Italian bred cultivar? What is its correct spelling (Sybilla or Sibilla)? Please use cultivar instead of variety.
In which phenological stage did you start the applications?
There is no statistical evalution on Fig. 8, but the results are important and very promising.
Please complete the Table 3 with marking the homogenity groups on it.
The conclusions are missing.
It is easy to understand.
Author Response
Response to Reviewer 2
Dear Reviewer,
Thank you for your comments and suggests concerning our manuscript entitled “Evaluation of strawberry fruit quality in terms of foliar calcium fertilization” (plants-2455513). Those comments are all valuable and very helpful for revising and improving our paper, as well as the important guiding significance to our researches. We have studied comments carefully and have made correction which we hope meet with approval. Revised portion are marked in red in the paper. The main corrections in the paper and the responds to the reviewer’s comments are as following.
Point 1. I think the introduction should contain some data about the optimal Ca content of fruits and leaves.
Response 1: We have not added information on the optimal Ca content of fruits and leaves, because the available literature provides general information on the effects of excess or deficiency of calcium on the properties of strawberries. The fruits of calcium (Ca)-deficient plants were significantly deformed when they were still green and were still small when they were ripe. It was also shown that the Ca content per dry weight had to be higher than 1.1% in aboveground plant organs to increase strawberry dry weight production. High Ca fertilization resulted in lower fruit acidity, regardless of cultivar. High Ca content and over-fertilization contributed to visual loss of fruit quality after harvesting.
Point 2. The „2. Results” and the third sub-cheapter („3. Results”) marked with the same sub-title, Please change the „2. Results” to „4. Materials and methods”. Correct order of the sub-chapteres for MDPI Plants is abstract, introduction, results, methods and materials, conclutions, references. Please douple check order of sub-chapters in your paper.
Response 2: The order of the chapters has been changed, in accordance with the journal's editorial requirements.
Point 3. I think, aim of a paper is to provide data and results, which are repeatable, therefore it is very important to write the Materials and methods prefectly. So, it should be provided more detailed information about the orchard desing (e.g. Was it one row, or more rows in the platation?, How wide was the bed? Were the plants planted on burm? How was the density or spacing in the examined orchard?, When was the orchard established? In which leaves were the plants? What kind of soil had the orchards? How was the Ca-supply of the soil?)
Response 3: Line 319-321 was added.
Point 4. In the strawberry production the cultivars change very frequently, therefore it would be good to add a very short description about Sybilla cultivar. Is this cultivar an Italian bred cultivar? What is its correct spelling (Sybilla or Sibilla)? Please use cultivar instead of variety.
Response 4: Line 313-318 was added: Strawberries "Sibilla" is a Italian, medium-ripening cultivar with high production potential. The fruits are bright red in color. This cultivar shows high adaptability both in the open field and under covers. 'Sibilla' is a cultivar suitable for the continental European climate.
Point 5. In which phenological stage did you start the applications?
Response 5: Line 321-322 was added: Fertilization treatments were carried out from the beginning of fruit ripening (81st phenological stage on the BBCH scale).
Point 6. Please complete the Table 3 with marking the homogenity groups on it.
Response 6: We supplemented Table 3 (now Table 2) and marked the homogeneity of the groups, indicating statistically significant differences between the control group and groups 1-8. (Page 5, and Line 151-152).
Point 7. The conclusions are missing.
Response 7: The Reviewer's suggestion was taken into account (line 442-455).

Reviewer 3 Report
This manuscript introduces an innovative tool designed for enhancing strawberry cultivation practices, focusing on the influence of tool setting conditions on fruit quality. The article aims to provide readers of the journal with valuable insights. However, there are several areas that require attention and improvement. The following suggestions are provided to assist the authors in revising their work:
1. Although the paper explores spraying conditions, the current title, "Evaluation of Strawberry Fruit Quality in terms of Foliar Calcium Fertilization," does not adequately reflect the central focus of the manuscript. Please consider revising the title to better align with the core content.
2. It would be helpful to clarify whether the Control group also received a calcium spray solution. Please provide an explanation in this regard.
3. Figures 6, 7, and 8 would benefit from a revised presentation format to effectively demonstrate the comparative differences between treatments. Consider utilizing a table format and incorporate symbols to indicate statistically significant differences. Additionally, in Figure 9, please indicate the results of the statistical analysis regarding the significance of the observed differences.
4. The paragraphs in Introduction (L58-L91) and Discussion (L333-L393) appear lengthy and could be further divided into smaller sections to enhance readability.
5. Based on the research findings, it is essential to clarify in the discussion and conclusion whether the observed improvement in strawberry fruit firmness is primarily attributed to foliar calcium spraying or spraying pressure. Please elaborate on the individual impacts of these aspects.
These suggestions aim to improve the overall clarity and presentation of the manuscript. I encourage the authors to address these points in their revision to enhance the quality of the article.
Author Response
Response to Reviewer 3
Dear Reviewer,
Thank you for your comments and suggests concerning our manuscript entitled Evaluation of strawberry fruit quality in terms of foliar calcium fertilization” (plants-2455513). Those comments are all valuable and very helpful for revising and improving our paper, as well as the important guiding significance to our researches. We have studied comments carefully and have made correction which we hope meet with approval. Revised portion are marked in red in the paper. The main corrections in the paper and the responds to the reviewer’s comments are as following.
Point 1. Although the paper explores spraying conditions, the current title, "Evaluation of Strawberry Fruit Quality in terms of Foliar Calcium Fertilization," does not adequately reflect the central focus of the manuscript. Please consider revising the title to better align with the core content.
Response 1: Title corrected to: (Effect of calcium foliar spray technique on mechanical properties of strawberries).
Point 2. It would be helpful to clarify whether the Control group also received a calcium spray solution. Please provide an explanation in this regard.
Response 2: The control group was not fertilized.
Point 3. Figures 6, 7, and 8 would benefit from a revised presentation format to effectively demonstrate the comparative differences between treatments. Consider utilizing a table format and incorporate symbols to indicate statistically significant differences. Additionally, in Figure 9, please indicate the results of the statistical analysis regarding the significance of the observed differences.
Response 3: We added information about analysis of variance in table 1. We have corrected Figure 9 (now Figure 4) by supplementing the percentage differences in firmness changes between groups 1-8 and the control group. The results of the statistical analysis regarding the significance of the observed differences are presented in the text (lines 153-166) and description of figure 4.
Point 4. The paragraphs in Introduction (L58-L91) and Discussion (L333-L393) appear lengthy and could be further divided into smaller sections to enhance readability.
Response 4: The reviewer's suggestion was taken into account (L58-94, L229-291).
Point 5. Based on the research findings, it is essential to clarify in the discussion and conclusion whether the observed improvement in strawberry fruit firmness is primarily attributed to foliar calcium spraying or spraying pressure. Please elaborate on the individual impacts of these aspects.
Response 5: The article shows that the improvement in fruit firmness was largely due to the adopted spraying technique (especially the lower speed and higher pressure of the treatment), while based on the degree of coverage, chemical and strength tests, it was proven that an effective increase in the value of measured parameters would not be possible without application of foliar spraying with a calcium agent (line 305-309).

Round 2
Reviewer 2 Report
The manuscript improved a lot, the authors completed it with a lot of missing parts, therefore I suggest to accept it for publication.
Its English improved a lot.
Reviewer 3 Report
The authors have made appropriate corrections as suggested and the revised manuscript is acceptable.